# Mannose-Binding Lectins as Potent Antivirals against SARS-CoV-2

**DOI:** 10.3390/v15091886

**Published:** 2023-09-06

**Authors:** Victória Riquena Grosche, Leandro Peixoto Ferreira Souza, Giulia Magalhães Ferreira, Marco Guevara-Vega, Tamara Carvalho, Romério Rodrigues dos Santos Silva, Karla Lilian Rodrigues Batista, Rodrigo Paolo Flores Abuna, João Santana Silva, Marília de Freitas Calmon, Paula Rahal, Luis Cláudio Nascimento da Silva, Bruno Silva Andrade, Claudener Souza Teixeira, Robinson Sabino-Silva, Ana Carolina Gomes Jardim

**Affiliations:** 1Laboratory of Antiviral Research, Institute of Biomedical Science (ICBIM), Federal University of Uberlândia (UFU), Uberlândia 38405-317, Brazil; victoriagrosche@gmail.com (V.R.G.); giumferreira95@gmail.com (G.M.F.); 2Institute of Biosciences, Languages, and Exact Sciences (Ibilce), São Paulo State University (Unesp), São José do Rio Preto 15054-000, Brazil; tamara@wcarvalho.com (T.C.); macal131@gmail.com (M.d.F.C.); rahalp@yahoo.com.br (P.R.); 3Innovation Center in Salivary Diagnostic and Nanobiotechnology, Institute of Biomedical Science (ICBIM), Federal University of Uberlândia (UFU), Uberlândia 38405-317, Brazil; leandro_peixototkd@hotmail.com (L.P.F.S.); marco.guevara.vega@gmail.com (M.G.-V.); 4Center of Agrarian Science and Biodiversity, Federal University of Cariri (UFCA), Crato 63130-025, Brazil; romeriorodrigues95@hotmail.com (R.R.d.S.S.); claudener@gmail.com (C.S.T.); 5Laboratory of Microbial Pathogenesis, CEUMA University, São Luís 65045-380, Brazil; biologakarlabatista@gmail.com (K.L.R.B.); luiscn.silva@ceuma.br (L.C.N.d.S.); 6Department of Biochemistry and Immunology, Ribeirão Preto Medical School, University of São Paulo, Ribeirão Preto 14049-900, Brazil; rodri_abuna@hotmail.com (R.P.F.A.); jsdsilva@fmrp.usp.br (J.S.S.); 7Oswaldo Cruz Foundation (Fiocruz), Bi-Institutional Platform for Translational Medicine, Ribeirão Preto 14049-900, Brazil; 8Laboratory of Bioinformatics and Computational Chemistry, State University of Southwest of Bahia, Jequié 45205-490, Brazil; bandrade@uesb.edu.br

**Keywords:** SARS-CoV-2, COVID-19, mannose-biding lectins, glycans, natural compounds, antivirals

## Abstract

The SARS-CoV-2 entry into host cells is mainly mediated by the interactions between the viral spike protein (S) and the ACE-2 cell receptor, which are highly glycosylated. Therefore, carbohydrate binding agents may represent potential candidates to abrogate virus infection. Here, we evaluated the in vitro anti-SARS-CoV-2 activity of two mannose-binding lectins isolated from the Brazilian plants *Canavalia brasiliensis* and *Dioclea violacea* (ConBR and DVL). These lectins inhibited SARS-CoV-2 Wuhan-Hu-1 strain and variants Gamma and Omicron infections, with selectivity indexes (SI) of 7, 1.7, and 6.5, respectively for ConBR; and 25, 16.8, and 22.3, for DVL. ConBR and DVL inhibited over 95% of the early stages of the viral infection, with strong virucidal effect, and also protected cells from infection and presented post-entry inhibition. The presence of mannose resulted in the complete lack of anti-SARS-CoV-2 activity by ConBR and DVL, recovering virus titers. ATR-FTIR, molecular docking, and dynamic simulation between SARS-CoV-2 S and either lectins indicated molecular interactions with predicted binding energies of −85.4 and −72.0 Kcal/Mol, respectively. Our findings show that ConBR and DVL lectins possess strong activities against SARS-CoV-2, potentially by interacting with glycans and blocking virus entry into cells, representing potential candidates for the development of novel antiviral drugs.

## 1. Introduction

In late 2019, an acute respiratory disease (COVID-19) caused by the severe acute respiratory syndrome coronavirus 2 (SARS-CoV-2) emerged and spread rapidly worldwide [1]. The World Health Organization (WHO) declared COVID-19 as a pandemic in March 2020 [2], and, up to now, almost 770 million people have been infected with SARS-CoV-2, and more than 7 million deaths have been caused by COVID-19 [3]. Although the massive vaccination campaigns presented significant results on the ongoing outbreak, antiviral drugs with direct action against SARS-CoV-2 are still essential [4,5]. The first drug approved by the FDA for COVID-19 treatment was VEKLURY^®^, or Remdesivir, which was initially developed against the Ebola virus [6]. LAGEVRIO^®^ (Molnupiravir) and PAXLOVID^®^ (Nirmatrelvir + Ritonavir) were also repurposed for the treatment of COVID-19 [7,8,9].

SARS-CoV-2, from the *Coronaviridae* family, is an enveloped, single-stranded positive RNA virus [10]. The viral genome is translated into two main groups of proteins: structural and non-structural proteins [10,11]. Structural proteins include the spike protein (S), matrix protein (M), and envelope protein (E), whereas the non-structural include proteases and RNA-dependent RNA polymerase (RdRP) [12]. On the outer surface of coronaviruses, spike glycoproteins are projected as its homo-trimeric state, which represents a pivotal recognition site used by the virus for attachment to and subsequent entry into the host cells. It is a complex of two subunits: S1 and S2 [10,12,13] (Figure 1A). The S1 subunit consists of the receptor binding domain (RBD) and N-terminal domain (NTD), which binds to the cellular receptors on the host cell membrane [10,11]. The most well-characterized SARS-CoV-2 receptor is the angiotensin-converting enzyme 2 (ACE-2) [10] (Figure 1B). However, CD147 [14] and NRP1 [15] have also been identified as viral entry receptors [10]. Otherwise, the S2 subunit consists of a cytoplasmic tail (CT), a transmembrane domain (TM), heptad repeat 2 (HR2), connector domain (CD), central helix (CH), heptad repeat 1 (HR1), and fusion peptide (FP) [16,17] (Figure 1A). The S1/S2 cleavage site is at the border between the two subunits. It is cleaved by proteases in the host cells, mainly the transmembrane protease serine 2 (TMPRSS2) [18], which activates the S protein, leading to the fusion of the viral envelope with the cell membrane of the host cells [17,19,20].

Molecular analyses indicates the presence of 22 N-glycan sites per monomer in the S protein [21] (Figure 1C). This carbohydrate–viral protein structure is critical for the modulation of SARS-CoV-2 membrane fusion [22], and the RBD-ACE-2 interactions (binding affinity, association, and dissociation) are affected by the presence of N-linked glycans in the complex [17]. Glycans are also important to SARS-CoV-2 evasion of the host immune system, since they can shield S protein epitopes, interfering with neutralizing antibody recognition [23]. There are two sites on the SARS-CoV-2 S protein where the glycosylation is mainly oligomannose-type: N234 and N709 [21]. These N-Mannose glycan sites, representing the NTD and FP regions of the spike (Figure 1A), respectively, can regulate the conformational dynamics “up” and “down” of RBD, modulating the interaction with the cell host receptor [24].

Lectins are proteins with a carbohydrate-binding domain possessing reversible binding capability to specific sugar moieties in glycoproteins or glycolipids, as well as free monosaccharide and glycan structures [25]. These proteins can recognize and interact with specific glycans present on the surface of viruses and other intracellular microorganisms, and they can alter the potential of these microorganisms to infect a host cell [26]. The *Canavalia brasiliensis* lectin, called “ConBR”, was first purified in 1979 from seeds of *C. brasiliensis*, also called Brazilian jackbean [27] (Figure 2A). Likewise, the lectin “DVL”, purified from *Dioclea violacea* seeds, a plant abundant in Brazilian cerrado vegetation (tropical savanna ecoregion), exhibits mannose-binding specificity [28] (Figure 2B).

Lectins isolated from plants have been used as biotechnological tools against several infectious diseases [29], demonstrating potent antibacterial and antiviral effects [30,31,32,33]. In the antiviral context, lectins isolated from seeds of *Bauhinia variegata* (specificity for glucose and galactose) demonstrated activity against Coxsackievirus B3 and Rotavirus [34,35]; from *Momordica charantia* (specificity for galactose and N-Acetylgalactosamine) against HIV, HSV-1, H1N1, H3N2, and H5N1 [36,37,38,39]; from *Mucuna pruriens* (specificity for mannose) against HCV [40]; and from *Senna tora* (specificity for mannose and galactose) against SARS-CoV-2 [41,42]. Therefore, we hypothesized that the mannose-biding lectins ConBR and DVL could present antiviral activity against SARS-CoV-2, potentially by interacting with the S protein to abrogate virus cell entry. Here, we evaluated the anti-SARS-CoV-2 effects of mannose-biding lectins (ConBR and DVL) originating from Brazilian flora. First, the anti-SARS-CoV-2 in vitro activity of these lectins was screened using a VSV-SARS-CoV-2 pseudotyped virus model. Then, the antiviral activity of ConBR and DVL was validated against infectious SARS-CoV-2 Wuhan-Hu-1 (SARS-CoV-2_WT_) and the SARS-CoV-2 variants Omicron and Gamma.

## 2. Materials and Methods

### 2.1. Extraction and Purification of ConBR and DVL

The *C. brasiliensis* and *D. violacea* seeds were collected from plants located at Crato, Ceará, Brazil, and the ConBR and DVL lectins were purified as previously described [43,44]. Briefly, the seeds from *C. brasiliensis* and *D. violacea* were milled to a fine powder. Subsequently, 5 g of each powder was incubated in 50 mL of 150 mM NaCl at 25 °C under continuous stirring for 4 h. Afterwards, the solubilized proteins in the supernatant were separated by centrifugation at 10,000× *g* at 4 °C for 20 min. Then, ConBR and DVL purifications were carried out by affinity chromatography using a Sephadex-G50 column (Sigma, Saint Louis, MO, USA) (2 × 20 cm) equilibrated with 100 mM NaCl. After the unbound proteins were washed out with the same solution, ConBR or DVL was eluted from the column using 0.1 M glycine at pH 2.6. The collected fractions containing ConBR or DVL were then analyzed by sodium dodecyl sulphate-polyacrylamide gel electrophoresis (SDS-PAGE), as previously published by our research group [43,44].

### 2.2. Cell Culture

Vero E6 cells (kidney tissue derived from a normal adult African green monkey, ATCC E6) or A549 cells (human lung epithelial adenocarcinoma, ATCC CCL185) were cultured BY N\J Bin Dulbecco’s modified Eagle’s medium (DMEM; Sigma-Aldrich, Saint Louis, MO, USA) supplemented with 100 U/mL penicillin (Gibco Life Technologies, New York, NY, USA), 100 mg/mL streptomycin (Gibco Life Technologies), 1% (*v*/*v*) non-essential amino acids (Gibco Life Technologies), and 10% (*v*/*v*) fetal bovine serum (FBS; Hyclone, Logan, UT, USA) at 37 °C in a humidified 5% CO_2_ incubator [45].

### 2.3. Cell Viability

Cell viability was measured by the MTT [3-(4, 5-dimethylthiazol-2-yl)-2, 5-diphenyl tetrazolium bromide] (Sigma-Aldrich) method. Vero E6 and A549 cells were seeded in a 96-well plate at a density of 1 × 10^4^ cells per well and incubated overnight at 37 °C in a humidified 5% CO_2_ incubator. Drug-containing media at concentrations of 50, 10, and 2 µq/mL were added to the cell culture. After 24 h at 37 °C, the media were removed and a solution containing MTT at the final concentration of 1 mg/mL was added to each well and incubated for 30 min at 37 °C in a humidified 5% CO_2_ incubator, after which the media were replaced with 100 μL of DMSO to solubilize the formazan crystals. Absorbance was measured by optical density (OD) of each well at 560 ηm, using the Glomax microplate reader (PROMEGA). Cell viability was calculated according to the equation (T/C) × 100%, where T and C represent the mean optical density of the treated group and vehicle control group, respectively. The cytotoxic concentrations of 50% (CC_50_) and 90% (CC_90_) were calculated using Graph Pad Prism 8.0 [46,47].

### 2.4. Virus Rescue and Titration

A pseudotyped vesicular stomatitis virus (VSV) expressing eGFP as a marker of infection, in which the glycoprotein gene (G) was replaced by the spike protein of SARS-CoV-2 (VSV-eGFP-SARS-CoV-2-S), was used for infection assays [48]. The virus was amplified employing Vero E6 cells in 175 cm^2^ flask. To determine viral titers, 1 × 10^4^ Vero E6 cells were seeded in each of 96 wells plate 24 h prior to the infection. Cells were infected with 10-fold serial dilutions of VSV-eGFP-SARS-CoV-2-S and supplemented with a medium containing 1% penicillin, 1% streptomycin, 1% non-essential amino acids, and 2% FBS. Infected cells were incubated for 24 h in a humidified 5% CO_2_ incubator at 37 °C. The viral foci were counted using EVOs cell imaging systems fluorescence microscopy (Thermo Fisher Scientific, Waltham, MA, USA) by detecting eGFP expression to determine viral titers, which were expressed in focus formation unit per milliliters (ffu/mL) [48].

SARS-CoV-2 Wuhan-Hu-1 (SARS-CoV-2_WT_) and variants Gamma and Omicron were amplified employing Vero E6 and A549 cells in a 175 cm^2^ flask. To determine viral titers, 1 × 10^5^ Vero E6 and A549 cells were seeded in each of 24 wells plate 24 h prior to the infection. Cells were infected with 10-fold serial dilutions of SARS-CoV-2 or variants Gamma and Omicron and supplemented with a medium containing 1% penicillin, 1% streptomycin, 1% non-essential amino acids, and 2% FBS. Infected cells were incubated for 48 h in a humidified 5% CO_2_ incubator at 37 °C. After incubation, the medium was removed, and the cells were fixed and stained with 4% paraformaldehyde and 0.4% crystal violet staining solution in order to visualize the formation of foci resulting from the cytopathic effect due to the release of the viral particle. From the foci number, it was possible to determine the viral titers of the supernatant in pfu/mL [49].

### 2.5. Antiviral Assays with VSV-SARS-CoV-2 and SARS-CoV-2_WT_

VSV-eGFP-SARS-CoV-2-S assays were carried out at a multiplicity of infection (MOI) of 0.005. For this, Vero E6 cells were seeded at a density of 1 × 10^4^ cells per well into 96-well plates 24 h prior to the infection. VSV-eGFP-SARS-CoV-2-S and the substance at non-toxic concentration were incubated for 1 h at room temperature, prior to the infection of cells with the inoculum for 2 h at 37 °C. The supernatant was removed, the cell monolayers were gently washed with 100 μL PBS, and the wells were completed with DMEM 2%. At 24 h post-infection (h.p.i.), the assays were analyzed using EVOs cell imaging systems fluorescence microscopy (Thermo Fisher Scientific) and the foci of infection were counted. The antiviral activity was calculated according to the equation (T/C) × 100%, where T and C represent the mean of the treated group and mean of the last concentration, respectively [48]. To assess the effective concentration of 50% (EC_50_) and 90% (EC_90_) of each lectin with the VSV-SARS-CoV-2 system, cells were infected with VSV-S and lectins at concentrations ranging from 200 µg/mL to 0.10 µg/mL for ConBR and 10 µg/mL to 0.005 µg/mL for DVL using the same protocol of antiviral assay. The EC_50_ and EC_90_ were calculated using GraphPad Prism software version 8.0.0. The values of CC_50_ and EC_50_ were used to calculate the selectivity index (SI = CC_50_/EC_50_) [47,50].

Vero E6 cells were seeded at a density of 2.5 × 10^4^ cells per well into 96-well plates 24 h prior to the infection. The infections of Vero cells with SARS-CoV-2_WT_ were performed at a multiplicity of infection (MOI) of 0.01 and lectins at non-toxic concentrations pre-determined on cell viability assays for 48 h. SARS-CoV-2_WT_ and lectins were incubated for 1 h at room temperature. Cells were infected with the inoculum at 37 °C for 1 h. The supernatant was removed, the cell monolayers were gently washed with PBS, and the wells were completed with DMEM 2%. The infection rates were determined at 48 h.p.i. by measuring cell death due to the infection using cellular viability assay. The antiviral effects were calculated according to the equation:$$\left\{\frac{\left[{\left({OD}_{T}\right)}_{SCoV2}–\Sigma {\left({OD}_{Infec.}\right)}_{SCoV2}\right]}{\left[\Sigma {\left({OD}_{Ctrl.}\right)}_{Cell}–\Sigma {\left({OD}_{Infec.}\right)}_{SCoV2}\right]}\right\}\times 100\%$$
where “(*OD*_T_)*_SCoV_*_2_” represents the optical density of the treated group infected with SARS-CoV-2; “Σ(*OD_Infec_*_._)*_SCoV_*_2_” represents the mean optical density of the infected group with SARS-CoV-2; and “Σ(*OD_Ctrl._*)*_Cell_*” represents the mean optical density of the cell control (non-infected group) [51].

### 2.6. Antiviral Assays with SARS-CoV-2_WT_ and Variants Gamma and Omicron Measured by Quantitative PCR

For the dose–response curves and all the time of drug-addition assays, Vero E6 or A549 cells were seeded at a density of 1 × 10^5^ cells per well into 24-well plates overnight, and infections were carried out with SARS-CoV-2_WT_ or variants Gamma and Omicron at an MOI of 0.01. The EC_50_ and EC_90_ of each lectin with SARS-CoV-2_WT_ or variants Gamma and Omicron were determined using the same protocol of VSV-eGFP-SARS-CoV-2-S, as previously described. In the drug-addition assays, lectins were administrated at non-toxic concentrations pre-determined on cell viability assays. In the pretreatment assay, cells were treated for 1 h at 37 °C with lectins, washed with PBS for compound removal, and then infected with the virus for 1 h at 37 °C. Then, cells were washed again to remove the unbound virus and replaced with a fresh medium for 24 h. In entry inhibition assays, cells were infected using a medium containing lectins and virus for 1 h at 37 °C, extensively washed with PBS, and incubated with a fresh medium for 24 h. The virucidal activity was evaluated using the same protocol of entry, with the exception of the inoculum containing the compound and virus incubated for 1 h prior to the addition to the cells. In the post-entry assay, Vero E6 and A549 cells were infected with SARS-CoV-2_WT_ for 1 h, washed with PBS, and immediately incubated in fresh media, with the addition of media containing compound at 4, 8, or 12 h.p.i. For all these protocols, virus titers were measured 24 h.p.i.

A general assay was performed with SARS-CoV-2_WT_ and its variants Omicron (HIAE-W.A; GISAID: EPI_ISL_6901961) and Gamma (IMT-MAN87209; GISAID: EPI_ISL_1060981). To this, Vero E6 and A549 cells were seeded at a density of 1 × 10^5^ cells per well into 24-well plates prior to the infection with virus at a multiplicity of infection (MOI) of 0.01 in the presence or absence of lectins at non-toxic concentrations for 24 h. The supernatant of all assays were collected and frozen for carrying out the following steps of extraction of viral RNA, cDNA synthesis, and real time PCR for viral titers quantification [52].

### 2.7. Mannose-Biding Lectins Blocking Assay

Vero E6 and A549 cells were seeded at a density of 1 × 10^5^ cells per well into 24-well plates 24 h prior to the infection. Cells were infected with SARS-CoV-2_WT_ at a multiplicity of infection (MOI) of 0.01 and treated with lectins at non-toxic concentrations in the presence or absence of D-(+)-Mannose (Sigma-Aldrich) at the final concentration of 0.1 M (mol/L). To evaluate a possible virucidal effect, inoculums containing compound and virus, in the presence or absence of D-(+)-Mannose (Sigma-Aldrich), were incubated for 1 h prior to the addition to the cells. The infection lasted 1 h at 37 °C, cells were extensively washed with PBS, and incubated with a fresh medium. At 24 h.p.i., the supernatant was collected and frozen for carrying out the following steps of extraction of viral RNA, cDNA synthesis, and real time PCR for viral titers quantification.

### 2.8. RNA Extraction and cDNA Synthesis

The Trizol-based RNA extraction protocol was adapted from Toni and collaborators [53]. For the volume of supernatant harvested from each well, a correspondent volume of Trizol (Invitrogen, Waltham, MA, USA) was added and homogenized. Then, chloroform (Merck KGaA, Darmstadt, Germany) was incubated with the samples for 3 min and centrifugated at 13,200 rpm for 15 min at 4 °C. The RNA-containing upper aqueous phase was collected, added to isopropanol (Merck KGaA) for 10 min at room temperature, and centrifugated at 13,200 rpm for 15 min at 4 °C. The supernatant was removed. The pellet was resuspended with ice-cold ethanol-depc 75% (Merck KGaA), centrifugated at 10,000 rpm for 15 min at 4 °C, and the supernatant was discarded. The pellet was resuspended in a final volume of 20 µL of nuclease-free water. After the quantification of RNA extracted, the samples should be frozen at −80 °C immediately [53].

Complementary DNA (cDNA) synthesis was performed using the High-Capacity cDNA Archive^®^ kit (Applied Biosystems, Waltham, MA, USA) according to the manufacturer’s instructions. The prepared reaction was composed of 10× RT Buffer, 25× dNTP Mix (100 mM), 10× RT Random Primers, MultiScribe™ Reverse Transcriptase, RNase Inhibitor, and Nuclease-free water, and the corresponding volume of viral RNA was extracted. The microtubes were submitted to the Veriti Thermal Cycler (Applied Biosystems^®^), incubated at 25 °C for 10 min, 37 °C for 120 min, and 85 °C for 5 min to generate complementary DNA in a reverse transcriptase-polymerase chain reaction.

### 2.9. Determination of Viral Load by Real-Time PCR

The RT-qPCR reactions were performed using Taqman Universal PCR master mix kit (Thermo Fisher Scientific, USA). The reactions consisted of 0.96 µM of each primer and 0.48 µM of probe specific to the viral nucleocapsid (N) gene (2019-nCoV_N1-F: 5′-GAC CCC AAA ATC AGC GAA AT-3′; 2019-nCoV_N1-R: 5′-TCT GGT TAC TGC CAG TTG AAT CTG-3′; and 2019-nCoV_N1-P: 5′-FAM-ACC CCG CAT TAC GTT TGG TGG ACC-BHQ1-3′) (Integrated DNA Technologies, Leuven, Belgium) [54], along with 5 µL of 2× Taqman Universal master mix (Thermo Fisher Scientific, USA) and 1.5 µL of cDNA in DEPEC ultrapure water. Reactions were performed using the following conditions: 94 °C for 10 min for enzyme activation; 45 cycles of denaturation at 95 °C for 15 s, annealing at 60 °C for 1 min. All amplifications were conducted on a QuantStudio 12 K Flex instrument (Applied Biosystems, Foster City, CA, USA). Serial tenfold dilutions of the standard plasmid of SARS-CoV-2 (2019-nCoV_N_Positive Control, 2 × 10^5^ genome copies/μL (gc/μL) (10^5^, 10^4^, 10^3^, 10^2^, 10^1^ and 10^0^), obtained from IDT (Integrated DNA Technologies, Leuven, Belgium), were used to produce standard curves to quantify the SARS-CoV-2 RNA copies. The limit of detection (LOD) parameters were similar to SARS-CoV-2_WT_ and the variants. Every RT-qPCR assay was performed in triplicate and included negative (nuclease-free water) and positive controls.

### 2.10. Protein Structures

The protein structures of ConBR (PDB: 3JU9) [55] with crystallographic resolution of 2.10 Å, DVL (PDB: 3AX4) [56] with 2.61 Å resolution, as well as the SARS-CoV-2 spike glycoprotein [17] with resolution of 2.80 Å were downloaded from the Protein Data Bank (https://www.rcsb.org/). For both structures of the lectins, we checked the amino acid clash and removed water molecules, crystallographic artifacts, and bound ligands using Pymol 2.3 program (Schrödinger 2023). In addition, we obtained a high-mannose molecule (PDB: 5VYB [57] with 2.40 Å, which was bound to the amino acid Asn234 of the SARS-CoV-2 spike protein for preparing this molecule for protein–protein docking; for this, we used Pymol 2.3 for binding this sugar molecule in this specific position [58]. 

### 2.11. Protein–Protein Docking

HADDOCK 2.4 [59] was used for performing docking calculations independently between ConBR and DVL structures with the modified SARS-CoV-2 spike protein. The docking process was directed to the modified spike protein residue Asn234 with the ConBR carbohydrate binding domain formed by the amino acids Leu99, Tyr100, Ser168, Ala207, Asp208, Thr226, and Arg228 [55] and the carbohydrate binding site from DVL formed by the amino acids Tyr12, Ans14, Leu99, Tyr100, Asp208, and Arg228 [56]. The five best docking clusters were analyzed considering their binding energies, as well as if the carbohydrate recognition binding sites from both proteins bound to the spike Asn234 modified amino acid region. For this, we used Pymol 2.3 for analyzing the interactions between protein chains [58].

### 2.12. Attenuated Total Reflection (ATR) Coupled to Fourier Transform Infrared (FTIR) Analysis

The samples were recorded in an ATR-FTIR spectrometer (Agilent Cary 630 FTIR, Agilent Technologies, Santa Clara, CA, USA). The diamond unit in the ATR platform performs an internal-reflection element to record the fingerprint infrared signature at the 1800 cm^−1^ to 800 cm^−1^ regions. The samples were prepared using lectins ConBR and DVL at 8 mg/mL and 10 mg/mL, respectively, and the VSV-eGFP-SARS-CoV-2-S at 1.4 × 10^7^ FFU/mL. A volume of 3 μL of each sample was inserted directly on the diamond cell and dehydrated to remove water functional groups for 10 min using airflow until each sample formed a thin dry layer on the ATR-crystal [60,61]. The spectra were then recorded (2 cm^−1^ resolution, 64 scans). The second derivative spectra were created based on original data plotted in the Origin Pro 9.8.0.200 (OriginLab, Northampton, MA, USA) software and adjusted using the Savitzky–Golay algorithm with polynomial order 2 and 20 points of the window [62,63].

### 2.13. Statistical Analysis

Individual experiments were performed in triplicate and all assays were performed a minimum of three times to confirm the reproducibility of the results. GraphPad Prism 8 software was used to assess statistical differences in the means of the readings using Student’s unpaired *t*-test or Mann–Whitney tests. *p* values < 0.05 were considered to be statistically significant.

## 3. Results

### 3.1. ConBR and DVL Block SARS-CoV-2 Entry to the Host Cells

Aiming to assess the potential of plant lectins to block virus entry, ConBR (from 200 µg/mL to 0.10 µg/mL) and DVL (from 10 µg/mL to 0.005 µg/mL) were first incubated with VSV-eGFP-SARS-CoV-2-S at an MOI of 0.005, and they were then used to infect Vero E6 cells for 2 h. Then, the inoculum was removed and replaced by fresh media. The effective concentration of 50% (EC_50_) and the cytotoxic concentration of 50% (CC_50_) were evaluated 24 h post-infection (h.p.i.), and values were calculated employing GraphPad Prism (Figure 3A). As a result, the treatment with both lectins strongly blocked VSV-eGFP-SARS-CoV-2-S infection, presenting CC_50_ of 2134 µg/mL, EC_50_ of 2.1 µg/mL, and a Selective Index (SI) of 1016.2 for ConBR (CC_90_ of 19,206.0 µg/mL, EC_90_ of 19.2 µg/mL) (Figure 3B), and the results for DVL were CC_50_ of 3.68 µg/mL, EC_50_ of 0.04 µg/mL, and an SI of 86.6 (CC_90_ of 33.12 µg/mL, EC_90_ of 0.36 µg/mL) (Figure 3C).

According to these data, the highest non-cytotoxic concentration (>80% cell viability) of each lectin was selected to evaluate the effects of ConBR and DVL in the context of the infection with SARS-CoV-2 Wuhan-Hu-1 (SARS-CoV-2_WT_). ConBR at 50 µg/mL or DVL at 2 µg/mL were incubated with SARS-CoV-2_WT_ for 1 h, and then were used to infect naïve Vero E6 cells for 1 h (MOI 0.01). Then, the inoculum was removed and replaced by fresh media. The inhibition of the infection was determined 48 h.p.i. by measuring cell death caused by the infection, using a cell viability assay (Figure 4A). The results demonstrated that ConBR and DVL significantly inhibited 39 and 36% of cell death resulting from SARS-CoV-2_WT_ infection, respectively, corroborating the antiviral activity of these lectins (Figure 4B).

### 3.2. ConBR and DVL Are Potent Inhibitors of SARS-CoV-2_WT_, but Also Variants Omicron and Gamma

The effects of the lectins in the context of the SARS-CoV-2_WT_ infection were further evaluated by assessing the replication rates in a dose–response assay, in a cell line derived from human lung epithelial adenocarcinoma (A549 cells), to better represent the infection in the human respiratory tract. The antiviral activity was also evaluated in the infection with Gamma and Omicron variants. A549 cells were treated with ConBR (from 200 µg/mL to 1.56 µg/mL) and DVL (from 10 µg/mL to 0.08 µg/mL) in the presence of SARS-CoV-2_WT_ or Gamma and Omicron variants at an MOI of 0.1 for 24 h, when replication levels were assessed. EC, CC, and SI were calculated. The results demonstrated that the lectins strongly inhibited SARS-CoV-2_WT_ and Gamma and Omicron variants infection (Figure 5). However, the antiviral potency of ConBR (Figure 5A) and DVL (Figure 5B) was higher against SARS-CoV-2_WT_ and Omicron than they were against Gamma infection. Values of EC_50_ and EC_90_, CC_50_ and CC_90_, and SI are shown in Figure 5C.

### 3.3. Multiple Effects of ConBR and DVL on the Replicative Cycle of SARS-CoV-2_WT_

To further investigate the antiviral activity of both lectins, time of drug-addition assays were performed to assess the effects of ConBR and DVL on different stages of the replicative cycle of SARS-CoV-2_WT_, and viral titers were quantified by measuring SARS-CoV-2_WT_ RNA levels in the supernatant of infected and/or treated cells. For all the time of drug-addition assays, Vero E6 cells were infected with SARS-CoV-2_WT_ at an MOI of 0.01 and virus titers were measured 24 h.p.i. In the pretreatment assay, cells were previously treated for 1 h with lectins at 37 °C, washed for compound removal, and then infected with the virus for 1 h at 37 °C. Then, cells were washed again to remove the unbound virus and replaced with a fresh medium (Figure 6A). Alternatively, for the entry inhibition assays, cells were infected using a medium containing lectins and virus for 1 h at 37 °C, extensively washed to the inoculum removal, and incubated with fresh medium (Figure 6B). The virucidal activity was evaluated using the same protocol of entry, except for the inoculum-containing compound, and the virus was incubated for 1 h prior to addition to the cells (Figure 6C). In the post-entry assay, cells were infected with SARS-CoV-2_WT_ for 1 h, washed to remove unbound virus, and incubated with a medium containing the compound at 37 °C (Figure 6D). All the supernatants were collected for viral titer quantification by RT-qPCR.

The results demonstrated that both lectins presented the highest rates of inhibition in the early stages of SARS-CoV-2_WT_ infection, mainly demonstrated in virucidal assay (Figure 6C). ConBR and DVL reduced viral titers in 1.25 × 10^5^ (97.4%) and 1.28 × 10^5^ (99.9%) RNA copies in the virucidal assay (Figure 6C), and in 1.23 × 10^5^ (95%) and 1.29 × 10^5^ (99.4%) in the entry assay (Figure 6B), respectively. Additionally, ConBR and DVL protected cells from infection at 88.4% (reduction of 2.3 × 10^5^ RNA copies) and 78.8% (reduction of 2 × 10^5^ RNA copies), respectively (Figure 6A), and presented post-entry inhibition at 88% (reduction of 2.5 × 10^5^ RNA copies) and 71.7% (reduction of 1.4 × 10^5^ RNA copies), respectively (Figure 6D)**.** Altogether, these data suggest the ConBR and DVL may act by different mechanisms of antiviral action, mainly affecting the early stages of the SARS-CoV-2_WT_ replicative cycle, potentially by interacting with virus particles.

To better understand if these lectins have any effect on late steps of the viral replication cycle, we performed an extended post-treatment assay (Figure 7). Vero E6 cells were infected with SARS-CoV-2_WT_ for 1 h, when supernatant was removed and replaced by fresh media. Treatments with ConBR and DVL were performed 4, 8, or 12 h.p.i., and virus titers were measured 24 h.p.i. (Figure 7A). Based on the results, the treatment with ConBR 4, 8 and 12 h.p.i. inhibited 79.1% (reduction of 1.3 × 10^6^ RNA copies), 98.7% (reduction of 1.6 × 10^6^ RNA copies) and 99% (reduction of 1.6 × 10^6^ RNA copies) of infection, respectively (Figure 7B). The same protocols of treatments with DVL resulted in viral inhibition of 79% (reduction of 1.3 × 10^6^ RNA copies), 99.1% (reduction of 1.6 × 10^6^ RNA copies) and 98.2% (reduction of 1.6 × 10^6^ RNA copies) (Figure 7C). As observed from these data, both lectins presented stronger effect from 8 h.p.i. (Figure 7B,C). Considering the evidence that SARS-CoV-2 virions are released, on average, after 12–36 h.p.i. [64], these data corroborate the effect of these lectins on viral particles that are being produced during the late stages of life cycle. Therefore, ConBR and DVL might affect both the entry of virus particles into naïve cells and the release of newly produced virions.

### 3.4. Interactions of ConBR and DVL with VSV-eGFP-SARS-CoV-2-S

The infrared spectra of VSV-eGFP-SARS-CoV-2-S, lectins ConBR or DVL, and VSV-eGFP-SARS-CoV-2-S plus lectins are represented in Figure 8. We found several molecular changes in VSV-eGFP-SARS-CoV-2-S incubated with ConBR or incubated with DVL. These changes in functional groups occurred in the biofingerprint region at 1800–800 cm^−1^, suggesting interactions between VSV-eGFP-SARS-CoV-2-S with ConBR (A, B, C, D, and E) and with DVL (F, G, H, I and J). As an outcome, the binding interactions between VSV-eGFP-SARS-CoV-2-S and ConBR suggested 17 vibrational modes (Figure 8A–E; Table 1) and 13 vibrational modes for VSV-eGFP-SARS-CoV-2-S plus DVL (Figure 8F–J; Table 1). We noticed that the mode vibrational at 1689 cm^−1^ and 1629 cm^−1^ (Figure 8A,F) were detected only after incubation in both lectins, suggesting interactions in the Amide I region (Table 1).

### 3.5. Insights on the Role of Mannose-Biding Lectins against SARS-CoV-2 Infections

The S protein of SARS-CoV-2 has several glycans rich in mannose residues that are essential for viral infection [24]. ConBR and DVL lectins are proteins that preferentially interact with glycans that have these carbohydrates in their structures. Therefore, to evaluate the involvement of the mannose–lectin interaction on SARS-CoV-2 infection, we performed an antiviral assay in the presence or absence of D-(+)-mannose, using Vero E6 and A549 cells, in order to evaluate its role on the antiviral activity of lectins ConBR and DVL, in a general infection and virucidal assays. To the general infection protocol, both cell lines were infected with SARS-CoV-2_WT_ (MOI = 0.01) and treated with lectins in the presence or absence of D-(+)-mannose at the final concentration of 1 M (mol/L) (Figure 9A). In the virucidal assay, lectins were previously incubated with the virus for 1 h in the presence or absence of D-(+)-mannose, at the same final concentration. The inoculums were then incubated with cells at 37 °C for 1 h. The supernatant was removed, and the cells were washed with PBS and replaced with DMEM 2% (Figure 9B). Supernatants were collected 24 h.p.i., and viral titers were quantified by RT-qPCR. As a result, the presence of mannose resulted in a lack of the antiviral activity by ConBR and DVL against SARS-CoV-2 infection in both cells, completely restoring virus titers (Figure 9C–F).

Additionally, in silico binding interactions between SARS-CoV-2 protein and both lectins were investigated, and the results indicated that ConBR interacted with the SARS-CoV-2 S protein by its carbohydrate-binding domain facing with the modified spike Asn234-high-mannose (Figure 10A). The HADDOCK docking cluster 2 presented the best binding and ConBR-spike complex interactions, with a predicted binding energy of −85.4 Kcal/Mol (±8.0) for its model 8. In addition, it is possible to verify, in this complex, that the high-mannose molecule interacted with the whole ConBR domain, suggesting the possible mechanism observed here in vitro. Furthermore, DVL carbohydrate binding site interactions with SARS-CoV-2 spike Asn234-high-mannose showed its best docking position in cluster 1 with a lower binding energy of −72.0 Kcal/Mol (±5.2) in comparison to ConBR docking (Figure 10B). On the other hand, it is possible to verify that at least the amino acids Leu99, Tyr100, and Asn14 are directly interacting with the high-mannose from the modified Asn234 of the spike protein, suggesting its possible mechanism of action against the SARS-CoV-2. The structural alignment between ConBR and DVL complexes with the modified Asn234 spike protein, which revealed that both lectins docked similarly with the viral glycoprotein, suggesting the same mechanism of action for both lectins on viral cell entry impairment (Figure 10C).

## 4. Discussion

Although a few antiviral drugs were approved by the FDA for COVID-19 treatment, the search for compounds with anti-SARS-CoV-2 activity is still a main global purpose. In this context, the lectins isolated from plants are proteins with broad applicability in biotechnology [78], and they can be used against different infections, such as protozoa, bacteria, and viruses [79,80,81]. These molecules act as carbohydrate-binding agents and are able to bind to glycans on the surface of SARS-CoV-2, altering the viral glycoprotein spike 3D conformation and tricking the interaction with the ACE2 cell receptor [82]. In this sense, the use of lectins has been hypothesized as a novel approach for the treatment of SARS-CoV-2 [81,82,83,84]. Here, we first reported the strong antiviral activity of the mannose-biding lectins ConBR and DVL against VSV-eGFP-SARS-CoV-2-S, SARS-CoV-2_WT_ Wuhan-Hu-1 strain, and Gamma and Omicron variants. 

By using VSV-eGFP-SARS-CoV-2-S, our results demonstrated that ConBR and DVL strongly inhibited the viral entry to the host cells, suggesting potential interactions of the lectins–spike glycoprotein, demonstrated by the high Selective Indexes (SI) of 1016.2 and 86.6, respectively. In the context of infection by SARS-CoV-2_WT_ and the variants Gamma and Omicron, ConBR presented SIs of 7, 1.7, and 6.5, respectively, and DVL presented SIs of 25, 16.8, and 22.3 against these SARS-CoV-2 variants. For these later assays, SIs were determined using a cell line derived from human lung epithelial adenocarcinoma (A549 cells) to better represent the infection in the human respiratory tract. Overall, the SI rates of both lectins were lower in infections with variant Gamma when compared to Omicron and SARS-CoV-2 Wuhan-Hu-1 strain infections. As reported by Spira, Gamma variant infections can display high transmissibility and a high degree of virulence [85]. On the other hand, McMahan and collaborators determined that Omicron infections decreased lung infectivity and pathogenic effects [86]. Wang and colleagues evaluated the anti-SARS-CoV-2 activity of 12 plant-derived lectins with different carbohydrate specificities and compared the infections of SARS-CoV, MERS-CoV, SARS-CoV-2 Wuhan-Hu-1, and variants Alpha, Beta, and Gamma [84]. According to their results, when compared with infection by the variants, the IC_50_ rates of the lectins against variant Gamma were slightly bigger than in the other variants, meaning that a higher concentration of the lectins was necessary to achieve the inhibition of 50% of the infection [84].

A previous work investigated the antiviral potential of lectins against coronaviruses and demonstrated that, from 14 mannose-specific agglutinins lectins, only a lectin derived from *Allium porrum* presented an SI > 100 [87]. In a more recent perspective, other mannose-biding lectins demonstrated anti-SARS-CoV-2 activity with SI rates lower than 100, as reported by Ahmed and collaborators [32]. Even though there is no antiviral specifically designed for SARS-CoV-2, Remdesivir was repurposed for the treatment of COVID-19 and, according to Choi and collaborators, in vitro assays using Vero E6 cells demonstrated an SI of 50 [88]. Cox and colleagues compared the activity of Remdesivir pro-drug GS-621763 against SARS-CoV-2 Wuhan-Hu-1 using VeroE6 or A549-ACE2 cells and obtained SIs of >137 and >51, respectively [89]. Franco and co-workers tested the effects of EIDD-1931, another licensed repurposed antiviral against SARS-CoV-2 (the active form of Molnupiravir), and demonstrated an SI of 12.5 against Omicron infection in A549-ACE2 cells [90]. These works represent a great reference for the comparison with our data, presented here, concerning the SI results of the treatments with ConBR and DVL.

The activities of ConBR and DVL were also investigated in different stages of the virus replicative cycle. Both lectins presented a strong inhibition of the early and late stages of viral infection, suggesting an effect of these lectins on viral particles, such as, for example, the virucidal action. A similar inhibitory effect was also observed for a lectin isolated from *Triticum vulgaris* (a type of “Wheat Germ Agglutinin”—WGA) against SARS-CoV-2 and its variants of concern Alpha and Beta [81]. In this study, Auth and colleagues demonstrated that WGA potently inhibits SARS-CoV-2 infection with an IC_50_ of <10 ng/mL, and it also had antiviral activity against variants Alpha and Beta. Using the same method we used for ConBR and DVL, Auth and co-workers indicated that WGA’s anti-SARS-CoV-2 activity was more effective upon preincubation of the lectin with the virus or when added during infection, and suggested that this lectin interacts with the spike glycoprotein, which is heavily glycosylated [21], blocking the interaction between viral glycoprotein and host cell receptors [81].

Since our data demonstrated a strong virucidal activity of ConBR and DVL, and based on the previously reported interactions of lectins–spike glycoprotein [21], we aimed to further investigate the role of the carbohydrate-binding site on the actions of these lectins against SARS-CoV-2 infection. Therefore, we performed a blocking assay using D-(+)-Mannose during the infection of SARS-CoV-2_WT_ of cells and treatment with ConBR and DVL. ConBR and DVL share the Carbohydrate Recognition Domain ligand (CRD-ligand) with mannose [28,78]; however, the anti-SARS-CoV-2 potential activity of these Brazilian lectins has not been explored yet. According to our data, the presence of mannose completely abrogated the anti-SARS-CoV-2 activity of these lectins and had similar results in both general infection and virucidal assays. 

At the beginning of the COVID-19 pandemic, Watanabe and collaborators revealed, through spectrometric analyses, the presence of mannose-glycan types associated with the spike glycoprotein [21], which was, afterwards, confirmed by Lokhande and colleagues through the use of molecular docking and simulation studies [91]. Wang and collaborators used a pseudovirus-based neutralization assay to assess the antiviral activity of a lentil lectin isolated from the *Lens culinaris* plant, along with 11 other plant-derived lectins with different carbohydrate specificities [84]. According to their results, *Lens culinaris*-derived lentil lectin, which specifically binds to oligomannose-type glycans, had the best antiviral activity, showing an IC_50_ of 40 μg/mL using the VSV pseudovirus model [84]. Additionally, a mannose-biding lectin isolated from *Lablab purpureus* seeds (FRIL), which is structurally similar to the plant lectin ConA, also exhibits effects against SARS-CoV-2 in vitro and in vivo [82]. In this context, Cavada and coworkers showed that ConBR presents 99% similarity to ConA [78]. Therefore, our data suggest that SARS-CoV-2 infection inhibition may occur by lectin–spike interaction, especially in RBD mannose-glycans, leading to a decrease in RBD-ACE2 binding, and, consequently, interfering in membrane cell fusion. 

Gong and colleagues reported that recombinant SARS-CoV-2 glycoproteins expressed in human cells, or from native S protein, presented high occupancy with oligomannose glycan type at N234 of RBD [24]. A regular N-glycan occupancy on S protein subunits [92] could be an opportunity for mannose-biding lectins to directly block the engagement of the S protein to the receptor and inhibit viral infections of host cells. 

We further investigated the potential binding interactions between SARS-CoV-2 S and the mannose-biding lectins studied here. Our data demonstrated that ConBR and DVL interacted with the SARS-CoV-2 S protein (binding energy of −85.4 Kcal/Mol and −72.0 Kcal/Mol, respectively) through its carbohydrate binding domain facing with the modified spike Asn234-high-mannose, suggesting the possible mechanism observed here in vitro. Additionally, the structural alignment between ConBR and DVL complexes with the modified Asn234 spike protein revealed that both lectins docked similarly with the viral glycoprotein, suggesting the same mechanism of action for these lectins on viral cell entry impairment. Previously, Lokhande and collaborators claimed that banana-derived mannose-specific lectin (BanLec) was able to target N-glycans of the spike glycoproteins to neutralize SARS-CoV-2 infectivity, and showed a biding energy of −219.8 Kcal/Mol with the S–protein complex [91]. Chikhale and colleagues tested lectins isolated from the medicinal plant *Withania somnifera*, also called Indian ginseng [93], and molecular dynamics (MD) suggested that “Withanoside X”, one of the substances derived from *W. somnifera*, presented binding free energy of −89.42 Kcal/Mol. Their results have also shown this lectin's ability to inhibit SARS-CoV-2 host entry and replication [93]. The infrared analysis confirmed the interaction with amide I, derived from protein structure and also from carbohydrate binding, in both ConBR and DVL complexes with VSV-eGFP-SARS-CoV-2-S. 

Interestingly, ConBR and DVL impaired post-entry replication of SARS-CoV-2_WT_ and also protected cells from infection. Although the main mechanism of action of other mannose-binding lectins is in the early stages of viral infection, according to our data, ConBR and DVL also inhibited up to 88% of SARS-CoV-2_WT_ replication. The strong antiviral activity during the post-entry stages implies that both lectins act not only by preventing SARS-CoV-2_WT_ entry to the host cells. Barton and co-workers, in a study of anti-HIV activity of Griffithsin (GRFT), a red-alga-derived lectin, determined that it not only interferes with virus entry, but also inhibits the viral protein production [94]. They also proposed that GRFT inhibits viral replication of other viruses, such as the hepatitis C virus and the Japanese encephalitis virus [94,95,96,97]. Additionally, in our data, ConBR and DVL protected cells from infection at rates of up to 88%. Lan and colleagues demonstrated that lectins possess the ability to agglutinate erythrocytes without altering the carbohydrates properties, since they have a minimum of one non-catalytic domain that binds reversibly to specific cellular monosaccharides or oligosaccharides [29]. These interactions with cellular receptor glycans could prevent the interaction with the spike glycoprotein and, therefore, suggest a mechanism of how it protects the cell from viral infection. 

To further explore the effects of lectins on the late stage of the viral replication cycle, we performed an extended post-treatment assay using Vero E6 cells. In the 4 h.p.i. treatment, lectins prevented up to 79% of SARS-CoV-2_WT_ infection. In the 12 h.p.i. treatment in both cells, the lectins presented excellent inhibition rates, both over 98%. Considering the evidence that SARS-CoV-2 virions are released, on average, after 12–36 h.p.i. [64], these data corroborate the effect of these lectins on viral particles that are being produced during the late stages of life cycle. Therefore, ConBR and DVL might affect both the entry of virus particles into naïve cells and the release of newly produced virions. The presented findings, however, are lacking information about the effect of these lectins on the production of infectious viral progeny since our assays evaluated an incubation of 24 h post infection. We believe that these data would be fruitful in future works for assessing the full antiviral potential of ConBR and DVL.

Regarding the SARS-CoV-2 antiviral research, the use of three-dimensional organoids is emerging as a desirable approach for understanding the virus–host interactions and for identifying novel therapeutic agents. The benefits of using 3D cell culture models to study respiratory virus infections, including COVID-19, and to search for anti-SARS-CoV-2 agents using primary human epithelial respiratory cells have been reported [98,99]. In this context, our data on the anti-SARS-CoV-2 activity of ConBR and DVL lectins support the possibility of future assays using organoids to better represent the human respiratory tract in vitro.

Mitchell and co-workers claimed that antiviral lectins are extensively being pursued clinically as anti-HIV microbicides via mucosal administration in successful in vivo rodent models [94,100]. Also, the mucosal via has been used in treatments with anti-H1N1 lectins [101] and anti-HSV-2 lectins [102]. Against SARS-CoV, an intranasal administration of the lectin Griffithsin (GRFT), from red algae, was used in a mouse model of pulmonary infection, which prevented weight loss, improved lung histopathology, and reduced lung tissue virus titers [100,103,104]. The active agent GRFT was also formulated as a rectal microbicide gel to prevent viral entry of HIV types 1 and 2, as well as HSV-2 and HCV, and was successfully tested in in non-human primates [105]. Boger and collaborators recently reported a phase I clinical trial evaluating the application of a topical rectal douche product containing Q-Griffithsin (Q-GRFT), which effectively reduced HIV transmission and did not disrupt the epithelial border or alter CD4+ cell distribution in the human rectal mucosa [106]. Although there is little published data about pre-clinical antiviral-based lectin drugs trials, based on these examples, it is possible to suggest that lectins are safe and tolerable as possible antiviral compounds. Therefore, our in vitro ConBR and DVL anti-SARS-CoV-2 data could be useful for future in vivo assays.

## 5. Conclusions

In summary, to the best of our knowledge, this study provides the first evidence of the carbohydrate-binding antiviral action of ConBR and DVL. Our findings show that mannose-biding plant lectins possess great potential as antiviral compounds and could be useful as templates for the development of novel antiviral drugs against SARS-CoV-2.

## Figures and Tables

**Figure 1 viruses-15-01886-f001:**
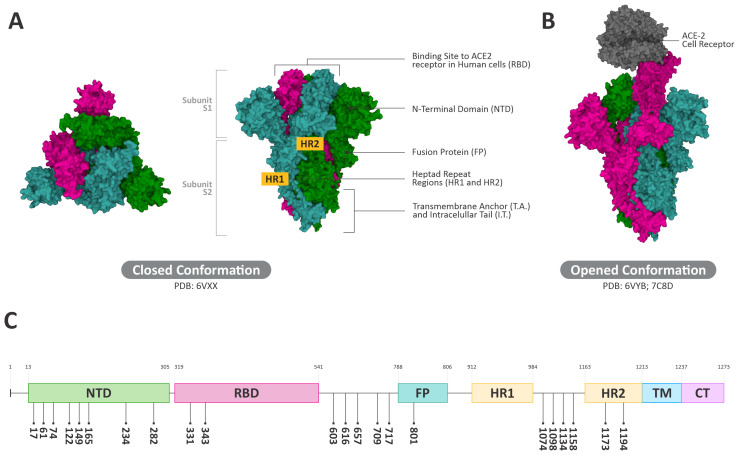
SARS-CoV-2 spike crystallographic structure as seen by electron microscopy, interactions with ACE-2 receptor, and schematic representation of the glycoprotein highlighting the positions of N-glycan sites. (**A**) Top view and side view of the spike glycoprotein at closed conformation (PDB ID: 6VXX). (**B**) Side view of the spike glycoprotein at closed conformation and its interaction with ACE-2 cellular receptor (PDB ID: 6VYB; 7C8D). (**C**) Schematic representation of the glycoprotein highlighting the positions of N-glycan sites per monomer.

**Figure 2 viruses-15-01886-f002:**
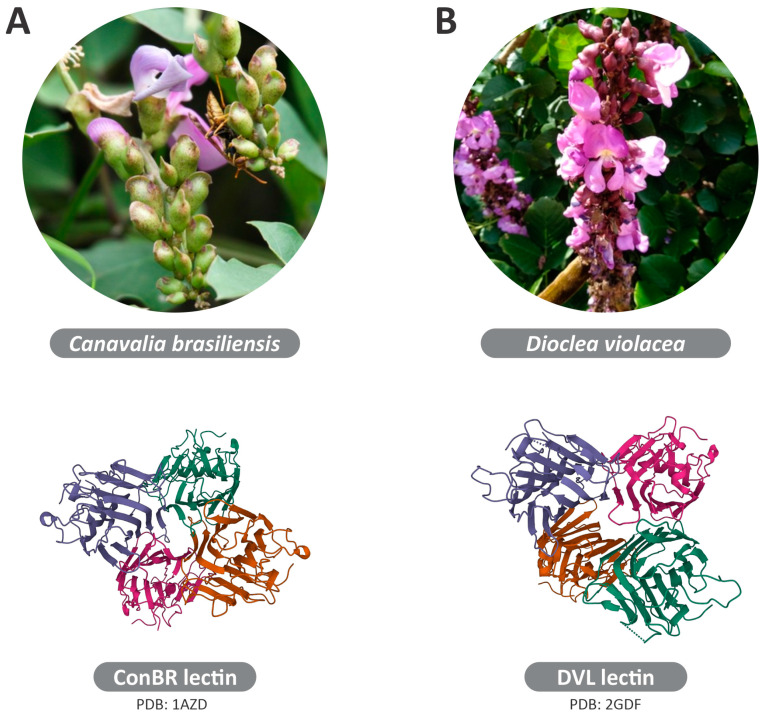
Lectins ConBR and DVL: Origin and 3D structure. (**A**) *Canavalia brasiliensis*, denominated “ConBR”, was first purified in 1979 from seeds of C. brasiliensis, also called Brazilian jackbean, and the 3D structure of the ConBR lectin (PDB ID: 1AZD). (**B**) Lectin purified from *Dioclea violacea* seeds, abundant in Brazilian vegetation, denominated “DVL”, exhibits mannose-binding specificity, and the 3D structure of DVL lectin (PDB ID: 2GDF).

**Figure 3 viruses-15-01886-f003:**
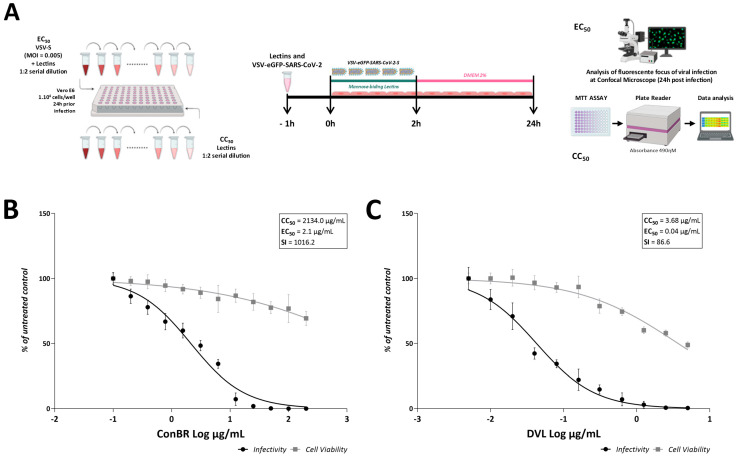
Lectins activity on infection of VSV-eGFP-SARS-CoV-2-S. (**A**) Schematic representation of the performed antiviral assay. Vero cells were infected with VSV-eGFP-SARS-CoV-2-S at an MOI of 0.005 and simultaneously treated with two-fold serial dilution of the compound, ranging from 200 µg/mL to 0.10 µg/mL (ConBR) (**B**) or 10 µg/mL to 0 µg/mL (DVL) (**C**). Viral infectivity and cell viability rates are indicated by black circles and gray squares, respectively. The effective concentration of 50% (EC_50_) and the cytotoxic concentration of 50% (CC_50_) were calculated employing GraphPad Prism. ConBR presented CC_50_ = 2134.0 µg/mL, EC_50_ = 2.1 µg/mL, and SI = 1016.2. DVL presented CC_50_ = 3.86 µg/mL, EC_50_ = 0.04 µg/mL, and SI = 86.6. Mean values are from at least three independent experiments, each measured in quadruplicate. Standard deviation is shown. Images and statistics analysis were generated using GraphPad Prism 8.

**Figure 4 viruses-15-01886-f004:**
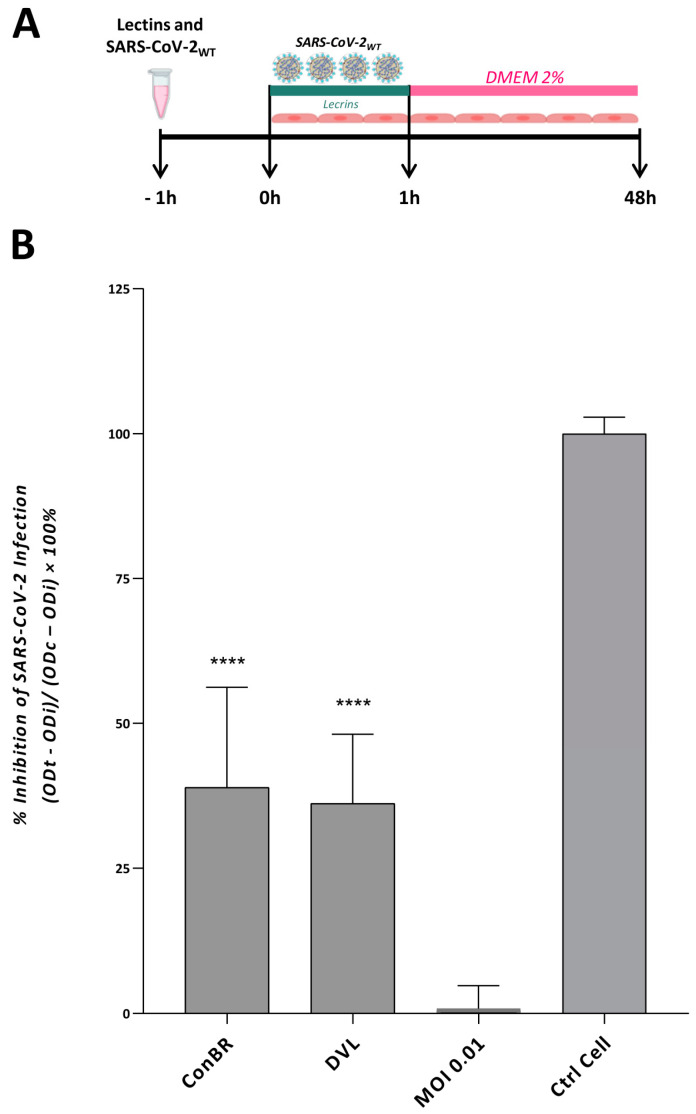
Effect of isolated lectins on cell death caused by SARS-CoV-2 Wuhan-Hu-1 strain infection. (**A**) SARS-CoV-2 and lectins were incubated for 1 h, and the inoculum was incubated with cells at 37 °C for 1 h. The supernatant was removed, and the cells were washed with PBS and replaced with DMEM 2%. At 48 h post-infection (h.p.i.), all the supernatants were removed, and the assays were analyzed according to the cell viability and the antiviral effect of each isolated lectin on reducing cell death. (**B**) A one-way ANOVA was performed to compare the effect of each lectin on SARS-CoV-2 Wuhan-Hu-1 strain infection. Mean values of three independent experiment each measured in triplicate including the standard deviations are shown. *p* values < 0.05 were considered significant. (****) *p* < 0.0001.

**Figure 5 viruses-15-01886-f005:**
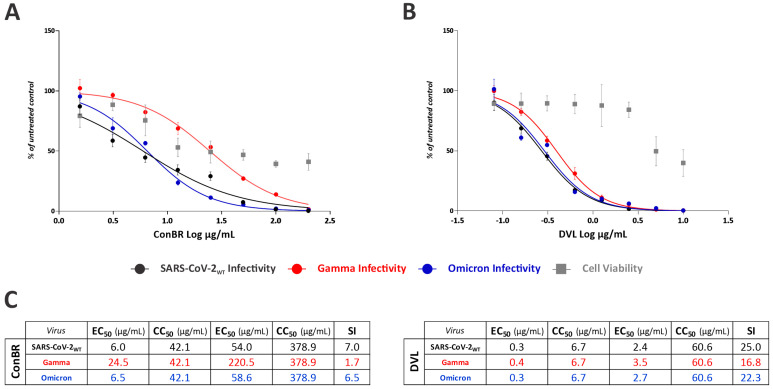
Lectins activity on infection by SARS-CoV-2_WT_ and Gamma and Omicron variants. A549 cells were infected with SARS-CoV-2_WT_, Gamma, and Omicron at an MOI of 0.1 and simultaneously treated with two-fold serial dilution of the compound, ranging from 200 µg/mL to 1.56 µg/mL (ConBR) or 10 µg/mL to 0.08 µg/mL (DVL). Viral infectivity rates are indicated by circles: black for SARS-CoV-2_WT_, red for Gamma, and blue for Omicron. Cell viability rates are indicated by gray squares. The effective concentration of 50% (EC_50_) and the cytotoxic concentration of 50% (CC_50_) were calculated employing GraphPad Prism. Cell viability was calculated using an MTT assay. (**A**) ConBR and SARS-CoV-2_WT_, Gamma, and Omicron infections. (**B**) DVL and SARS-CoV-2WT, Gamma, and Omicron infections. Mean values are from at least three independent experiments, each measured in quadruplicate. Standard deviation is shown. Images and statistics analysis were generated using GraphPad Prism 8. (**C**) Table summarizing EC_50_, CC_50_, EC_90_, CC_90_, and SI rates of both lectins and each variant.

**Figure 6 viruses-15-01886-f006:**
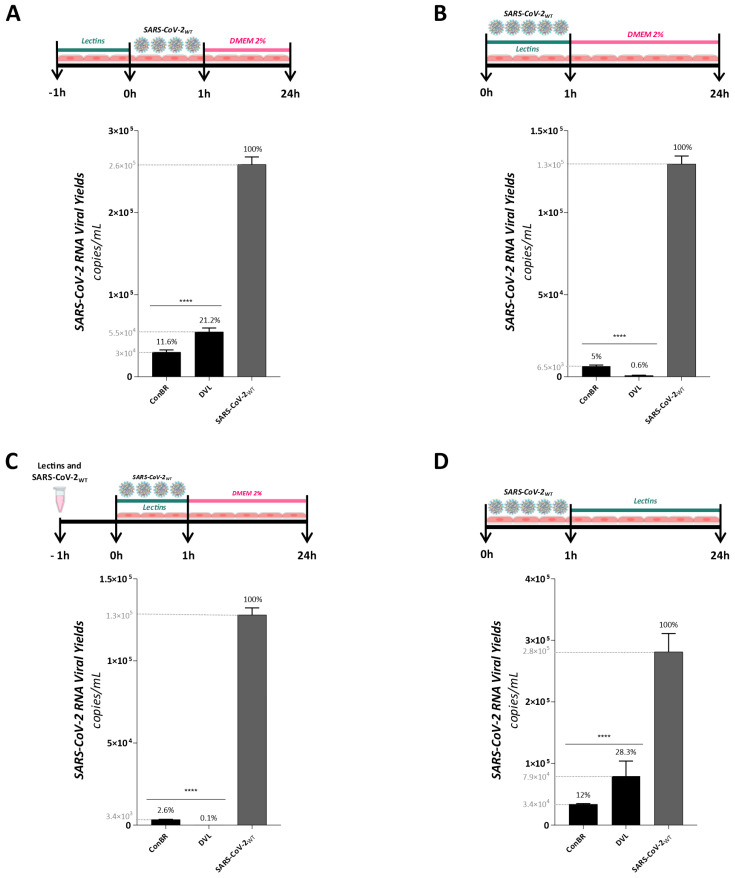
Effect of isolated lectins on different stages of SARS-CoV-2 Wuhan-Hu-1 strain replicative cycle. (**A**) Pre-treatment assay: cells were incubated with the lectins for 1 h. Media was removed, and cells were infected with SARS-CoV-2 for 1 h at 37 °C. The supernatant was removed, and cells were washed with PBS and replaced with DMEM 2%; (**B**) Entry assay: cells were infected with SARS-CoV-2 and simultaneously treated with lectins for 1 h. The supernatant was removed, and cells were washed with PBS and replaced with DMEM 2%; (**C**) Virucidal assay: SARS-CoV-2 and lectins were incubated for 1 h, and then the inoculum was incubated with cells at 37 °C for 1 h. The supernatant was removed, and cells were washed with PBS and replaced with DMEM 2%; (**D**) Post-entry assay: cells were infected with SARS-CoV-2 for 1 h. The supernatant was removed, cells were washed with PBS, and cells were treated with each lectin for 24 h. For all antiviral protocols, supernatants were collected 24 h post-infection (h.p.i.); viral RNA were extracted; complementary DNA was synthesized; and a real time PCR for viral titer quantification was performed. Mean values of three independent experiment, each measured in triplicate, including the standard deviation are shown. *p* values < 0.05 were considered as statistically significant. (****) *p* < 0.0001.

**Figure 7 viruses-15-01886-f007:**
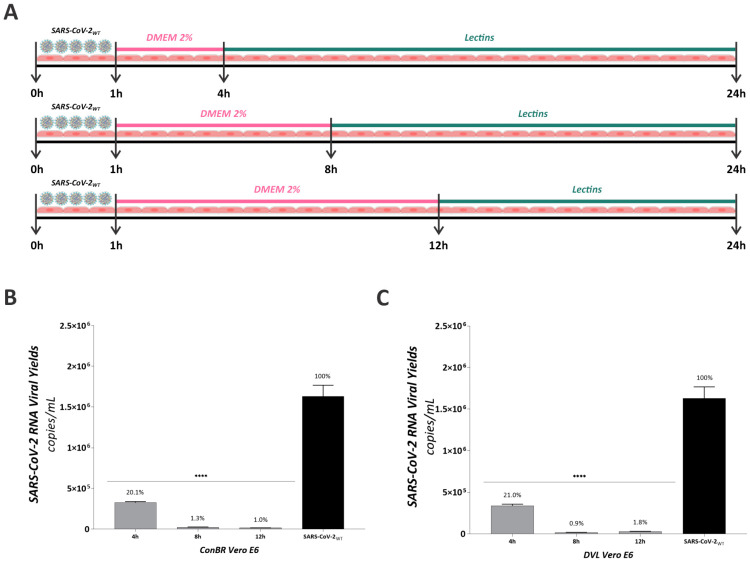
Effect of lectins on SARS-CoV-2 Wuhan-Hu-1 infectivity at different times of post-entry assays. (**A**) Schematic representation of this antiviral assay. Vero E6 cells were infected with SARS-CoV-2 Wuhan-Hu-1 virus for 1 h, the supernatant was removed, and cells were washed with PBS and replaced with fresh media. Treatments with ConBR at 50 µg/mL (**B**) and DVL at 2 µg/mL (**C**) were performed 4, 8, or 12 h.p.i., and virus titers were measured 24 h.p.i. by q-RT-PCR. A one-way ANOVA was performed to compare the effect of each lectin on SARS-CoV-2 Wuhan-Hu-1 infection. Mean values of three independent experiment, each measured in triplicate, including the standard deviation are shown. *p* values < 0.05 were considered significant. (****) *p* < 0.0001.

**Figure 8 viruses-15-01886-f008:**
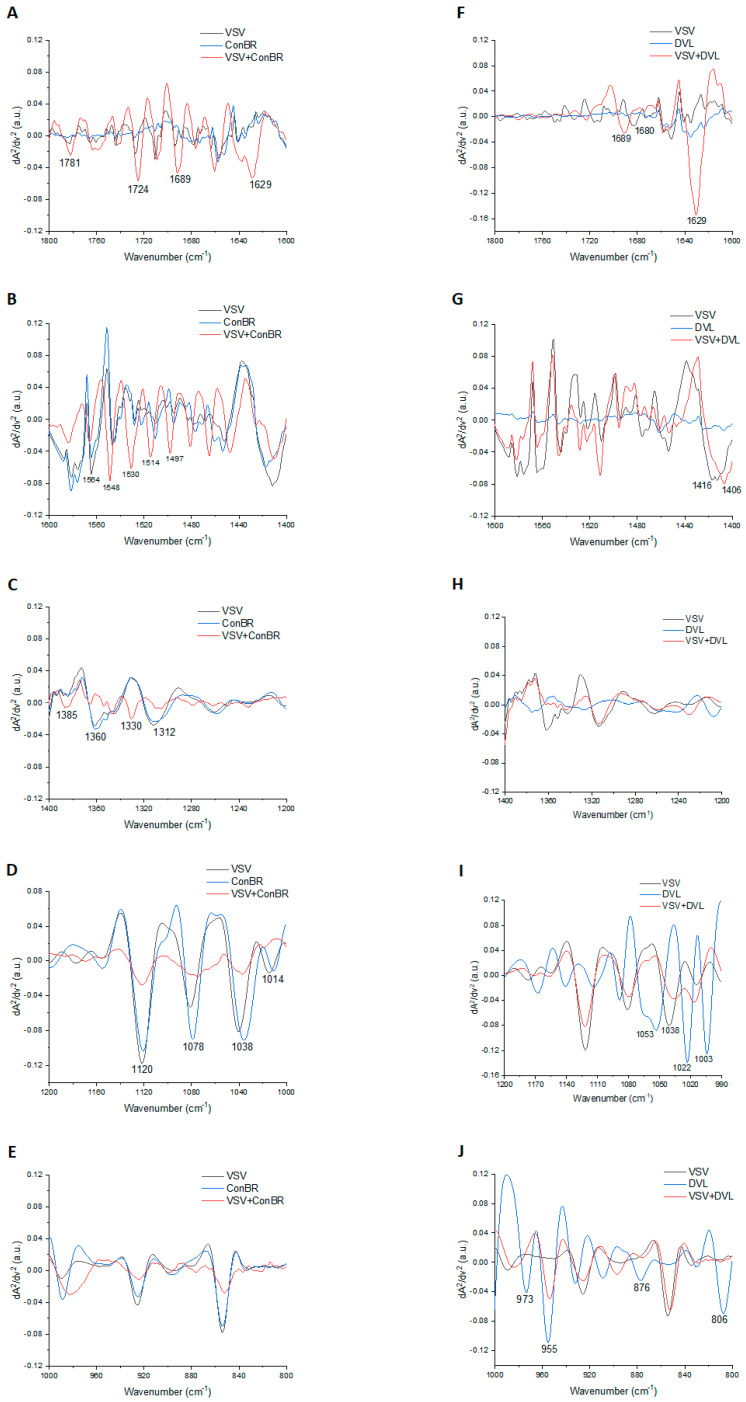
Infrared spectroscopy indicates molecular interactions between VSV-eGFP-SARS-CoV-2-S plus ConBR and VSV-eGFP-SARS-CoV-2-S plus DVL. Representative scheme of the ATR-FTIR technology with VSV-eGFP-SARS-CoV-2-S (black line), ConBR or DVL (blue line), and VSV-eGFP-SARS-CoV-2-S incubated with ConBR or DVL (red line). The VSV-eGFP-SARS-CoV-2-S and ConBR spectra are depicted in panels (**A**–**E**) and VSV-eGFP-SARS-CoV-2-S and DVL spectra are depicted in panels (**F**–**J**).

**Figure 9 viruses-15-01886-f009:**
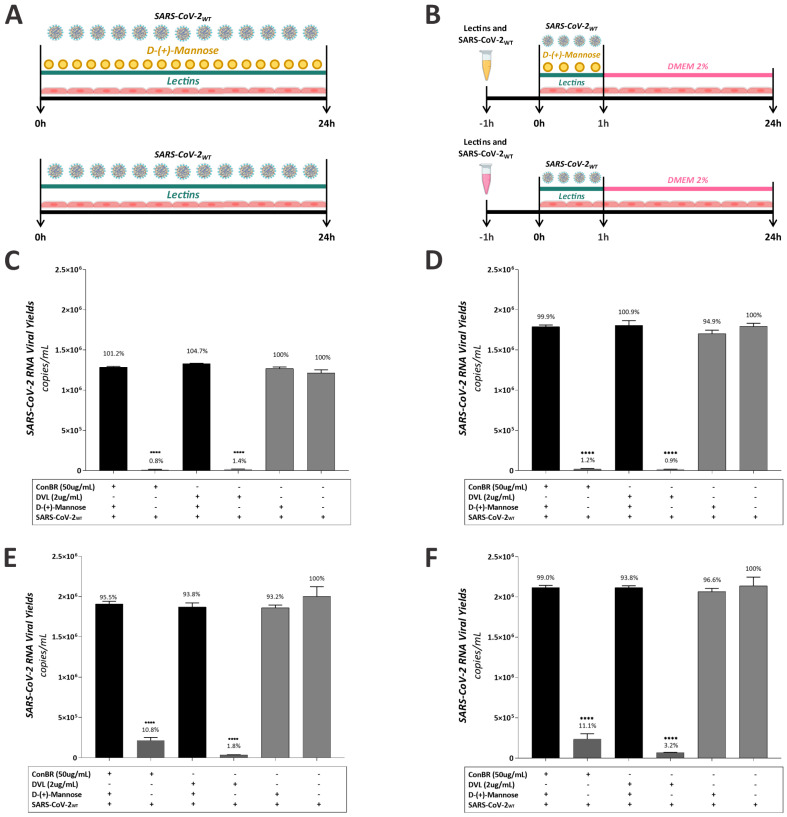
Effect of lectins on SARS-CoV-2 Wuhan-Hu-1 in the presence of D-(+)-Mannose. (**A**) Schematic representation of general infection antiviral assay. (**B**) Schematic representation of virucidal assay. Vero E6 (**C**) and A549 cells (**E**) were infected with SARS-CoV-2 Wuhan-Hu-1 virus and treated with ConBR at 50 µg/mL and DVL at 2 µg/mL, in the presence or absence of D-(+)-Mannose. SARS-CoV-2 and lectins were incubated for 1 h, and the inoculum was incubated with Vero E6 (**D**) and A549 cells (**F**) at 37 °C for 1 h. The supernatant was removed, and cells were washed with PBS and replaced with DMEM 2%. For all protocols, the supernatants were collected 24 h.p.i.; viral data were quantified by p-RT-PCR. A one-way ANOVA was performed to compare the effect of each lectin in the presence or absence of D-(+)-Mannose on SARS-CoV-2 Wuhan-Hu-1 infection. Mean values of three independent experiment, each measured in triplicate, including the standard deviation are shown. *p* values < 0.05 were considered significant. (****) *p* < 0.0001.

**Figure 10 viruses-15-01886-f010:**
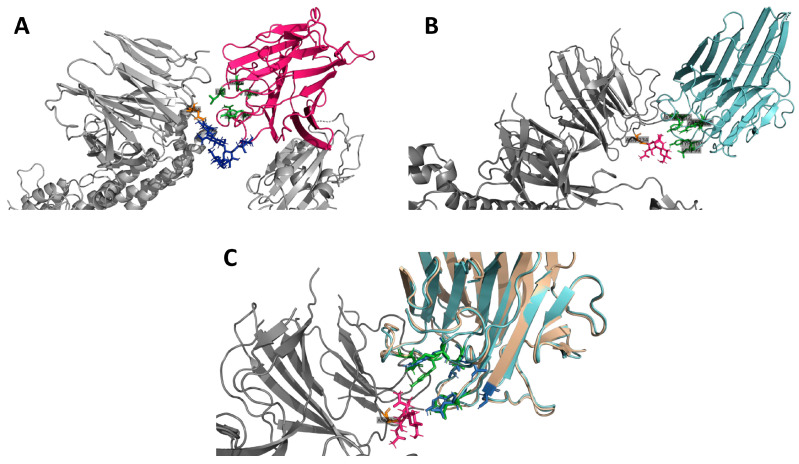
Protein docking between SARS-CoV-2 Spike glycoprotein and Lectins. (**A**) Protein docking between SARS-CoV-2 spike glycoprotein (grey) and ConBR (hot pink). The spike active amino acid Asn234 (orange) is bound to a high-mannose molecule (dark blue), and a docking interface is formed with the ConBR carbohydrate domain (green). (**B**) Protein docking between SARS-CoV-2 spike glycoprotein (grey) and DVL (Cyan). The spike active amino acid Asn234 (orange) is bound to a high-mannose molecule (hot pink), and a docking interface is formed with the DVL carbohydrate binding site formed by the amino acids Tyr12, Ans14, Leu99, Tyr100, Asp208, and Arg228 (green). (**C**) The 3D structural alignment between both complexes formed by ConBR (wheat) and DVL (cyan) with the SARS-CoV-2 spike glycoprotein (grey). The spike active amino acid Asn234 (orange) is bound to a high-mannose molecule (hot pink). Both ConBR carbohydrate binding domain (marine blue) and DVL carbohydrate binding site (green) are aligned and positioned similarly to the viral protein.

**Table 1 viruses-15-01886-t001:** Tentative assignments for vibrational modes indicating interaction in VSV-eGFP-SARS-CoV-2-S with ConBR or DVL.

Lectins	Peak	Tentative Assignment	Reference
ConBR	1781	Carbonyl C=O ester stretching region	[65]
	1724	C=O stretching band mode of the fatty acid ester	[66]
	1689	Amide I (disordered structure-non-hydrogen bonded)	[67]
	1629	Amide I region	[68]
	1564	Ring base	[69]
	1548	Amide II	[70]
	1530	Stretching C=N, C=C	[69]
	1514	Amide II	[71]
	1497	C=C, deformation C-H	[69]
	1385	Deformation C-H	[69]
	1360	Deformation C-H	[69]
	1330	CH_2_ wagging	[70]
	1312	Amide III band components of proteins	[70]
	1120	Mannose-6-phosphate	[66]
	1078	Phosphate I in RNA	[72]
	1038	Stretching C-O ribose	[69]
	1012	Stretching C-O deoxyribose	[69]
DVL	1689	Amide I (disordered structure-non-hydrogen bonded)	[67]
	1680	Unordered random coils and turns of amide I	[73]
	1629	Amide I region	[68]
	1416	Deformation C-H, N-H, stretching C-N	[69]
	1406	CH3 asymmetric deformation	[74]
	1053	νC-O and δC-O of carbohydrates	[75]
	1038	Stretching C-O ribose	[69]
	1022	Glycogen	[68]
	1003	Carbohydrate residues attached to collagen and amide III vibration	[76]
	973	OCH3 (polysaccharides, pectin)	[67]
	955	Phospholipids/carbohydrates	[77]
	876	(A-form helix) conformation	[72]
	806	(A-form helix) conformation	[72]

## Data Availability

Not applicable.

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
