# Peer review of "Mannose-Binding Lectins as Potent Antivirals against SARS-CoV-2"

_viruses, 2023, doi:10.3390/v15091886_

Round 1

Reviewer 1 Report

The manuscript by Grosche et al is well written and thorough.  While the identified compounds are unlikely to be development candidates, the authors presented sufficient evidence to suggest that they do have antiviral activity and could be used as a starting point for a medicinal chemistry effort to increase potency and decrease cytotoxicity.  While the virology is interesting, if there is to be any real world utility of this data, it would helpful for the authors to include what is known about the safety and tolerability of these compounds (and related compounds) in humans.  

Reviewer 2 Report

MAIN CONCERNS

1) Glycan structures differ depending on the species and tissue. A a major limitation of this manuscript is that all experiments  were performed exclusively using Vero cells (African green monkey kidney cells). However, SARS-Cov-2 replicates in the epithelium of the human respiratory tract.

o   At least a subset of experiments must be repeated using human lung epithelium cell lines such as A549.  SARS-CoV-2 virus stocks and pseudotyped VSV-eGFP-SARS-CoV-2 virus stocks must be produced in a human lung epithelium cell line.

o   The gold standard would be multiple cycle infection assays using differentiated primary human lung epithelial cells cultured in 3-dimensional tissues on the air-liquid-interphase, starting with a very low MOI (for example, see 10.1016/j.antiviral.2021.105122). This would ensure that the virus is amplified in a biologically relevant cell system, ensuring that the viral S protein, which is suggested to be the target of the lectins in this paper, has the tissue and species-specific glycan structure. It is strongly suggested to repeat key experiments using this system.

o   Ensure that the glycosylation pattern of the S-protein expressed in Vero cells is representative of S protein expressed in human lung epithelial cells.

2) Through-out the manuscript, positive controls are missing. Well described positive controls allow the reader to obtain a better understanding of the inhibitory activity of the two lectins. Please see details below.

OTHER CONCERNS

-      Line 29: spell out ConBr and DVL

-       Lines 29-30: indicate the SARS-CoV-2 lineage of the S protein used; is “VSV-eGFP-SARS-CoV-2-S” identical to “VSV-eGFP-SARS-CoV-2” later in the manuscript?

-       Lines 50-51: incorrect. Paxlovid and Veklury are FDA approved, while Lagevrio received an EUA

-       Lines 113-114: please provide rational why these two lectins (and not other mannose-binding lectins) were chosen. Are human receptors known for the two lectins? Do the two lectins have mitogenic activity?

-       Lines 130-131: please describe the results: single band, size of band(s), etc?

-       Be consistent in using decimal points, not comma

-       Figure 3B and C: subscript EC50 and CC50 in legend

-       Lines 227-228: how were the SARS-CoV-2 VoCs Omicron and gamma produced? Which cell line?

-       Lines 328-331: A positive control is missing, which would allow to get a better understanding of the inhibitory activity of the lectins in the assay. Include a well-described positive control, ideally with a similar mechanism of action, such as one of the Spike targeting antibodies (https://www.ncbi.nlm.nih.gov/pmc/articles/PMC8702401/)

-       Lines 346-348: reductions in SARS-CoV-2 infection by 36-39% are not impressive and indicate that the majority of the cells are not protected from infection. A dose-response curve with a determination of the EC50 and EC90 as in Figure 3 would provide a better understanding of the antiviral potency of the lectins. A positive control is missing. Please include a well-described antiviral compound in the assay such as nirmatrelvir (https://pubmed.ncbi.nlm.nih.gov/34726479/) or an entry inhibitor (https://www.ncbi.nlm.nih.gov/pmc/articles/PMC8702401/)

-       Figure 5:

o   The cartoons depicting the exact order of addition are very informative and appreciated!

o   A variation of the experiment in 5D is suggested, where, after the 1h infection phase and washes, treatment with the lectins is delayed by several hours. This would inform if the lectins have any effect on a late step in the viral life cycle.

-       Figure 6:

o   What is the lower limit of quantification? Is it the same for Wuhan, Gamma and Omicron?

o   Dose-response curves would inform if the maximum activity of the two lectins against Gamma and Omicron has been reached.

-       Figure 8:

o   what is the molar ratio between mannose and ConBR and DVL, respectively? If mannose is in far excess over the lectin concentration, a dose-response titration is needed.

o   It would be very informative to repeat this experiment in the virucidal assay format

-       Lines 492-493. This statement is wrong. Please see above and correct.

-       Lines 511-515: these conclusions are partially wrong and over-simplified:

o   Chloroquine and Lopinavir have not been approved for the treatment of SARS-CoVB-2 and multiple clinical trials demonstrated that they were ineffective in the treatment of SARS-CoV-2

o   The in vitro SI of remdesivir depends on the exact cell system used.

o   High in vitro SI are not predictive of clinical efficacy

o   In vitro SIs can only be used to compare inhibitors with the same mechanism of action.

-       Lines 595-596: Please discuss the following aspects of antiviral drug development of lectins:

o   How would the lectins be administered?

o   SARS-CoV-2 replicates in the respiratory tract:

§  Is there a precedent of efficient delivery of lectins at estimated efficacious concentrations?

§  ConBr lectin has been shown to be mitogenic (https://link.springer.com/article/10.1007/s00441-011-1239-x). How would you address the expected inflammatory reaction when delivered to the lung?

Overall, the English language used is understandable. Moderate proof-reading would further improve the quality of the manuscript.

Round 2

Reviewer 2 Report

Improvements have been made to the manuscript; however, the main concerns raised in the original review have not been adequately addressed:

Point 1: The authors still rely heavily on the use of Vero cells, a kidney cell line isolated from African green monkeys and not representative of the human respiratory tract, where SARS-CoV-2 replicates. Only a subset of experiments were confirmed using the human lung epithelial cell line A549. However, virus stocks were still produced in Vero cells. Since glycan structures often differ depending on the species and tissue, the glycan structures on the viral S protein produced in Vero cells might be significantly different from viral S protein produced in a human lung epithelial cell line. A simple experiment was suggested to address the issue: Infection of human lung epithelial cells at a very low MOI (MOI < 0.001) and a sufficiently long incubation time to allow multiple cycles of infection (at least 96 hours). This would ensure that during the multiple rounds of infection, the vast majority of the virus is produced by human lung epithelial cells and is no longer derived from Vero cells and should therefore carry the appropriate glycan structure. While differentiated primary human lung epithelial cells in 3D culture might not be available, A549 or a similar human lung epithelial cell line using a traditional 2D culture system would be appropriate.

The reference to the work of Essaidi-Laziosi (https://pubmed.ncbi.nlm.nih.gov/34485956/) as a justification of using Vero cells is insufficient and irrelevant. Essaidi-Laziosi compared Vero and primary airway epithelial cells for the purpose of virus detection. However, they did not use Vero cells to address the effect of lectins on viral s protein and/or the host ACE2 receptor. In fact, they even point out that Vero cells " do not mimic the primary site of entry in the human respiratory tract".

Point 2: Positive controls are still missing. The author’s opinion that approved direct antivirals such as remdesivir, molnupiravir and nirmatrelvir are repurposed from other viral diseases is irrelevant and, in the case of nirmatrelvir, also incorrect. Referencing these drugs in the discussion is not sufficient. The point of a positive control is to validate experiments and to give readers additional perspective of the antiviral activity of the lectins in comparison to the positive control. A positive control such as nirmatrelvir can easily be incorporated in the experiment suggested above (A549 cells, low MOI, incubation time > 96h).

Points 3 and 4 have been adequately addressed.

Point 5: Nirmatrelvir, the active ingredient of Paxlovid, was specifically developed for the treatment of SARS-CoV-2 infection. Reference 8 does not mention that it was originally repurposed from other viral infections such as Ebola or Chikungunya. Please correct.

Points 6-8 have been adequately addressed. Please include all relevant discussions and explanations provided in the rebuttal letter also in the manuscript.

Point 9: A positive control is still missing; please see above.

Points 10-16 have been adequately addressed. Please include all relevant discussions and explanations provided in the rebuttal letter also in the manuscript.

Important: Please double-check the graphs in Figures 6A and B. They appear identical. 6B in the current version is different from Figure 6B in the original manuscript.

Some minor editing of the manuscript, especially of the discussion, would further improve its quality.
